# Mitochondrial Transplantation in Mitochondrial Medicine: Current Challenges and Future Perspectives

**DOI:** 10.3390/ijms24031969

**Published:** 2023-01-19

**Authors:** Marco D’Amato, Francesca Morra, Ivano Di Meo, Valeria Tiranti

**Affiliations:** Unit of Medical Genetics and Neurogenetics, Fondazione IRCCS Istituto Neurologico Carlo Besta, 20126 Milan, Italy

**Keywords:** mitochondria, mitochondrial diseases, mitochondrial medicine, mitochondrial dysfunction, mitochondrial transplantation

## Abstract

Mitochondrial diseases (MDs) are inherited genetic conditions characterized by pathogenic mutations in nuclear DNA (nDNA) or mitochondrial DNA (mtDNA). Current therapies are still far from being fully effective and from covering the broad spectrum of mutations in mtDNA. For example, unlike heteroplasmic conditions, MDs caused by homoplasmic mtDNA mutations do not yet benefit from advances in molecular approaches. An attractive method of providing dysfunctional cells and/or tissues with healthy mitochondria is mitochondrial transplantation. In this review, we discuss what is known about intercellular transfer of mitochondria and the methods used to transfer mitochondria both in vitro and in vivo, and we provide an outlook on future therapeutic applications. Overall, the transfer of healthy mitochondria containing wild-type mtDNA copies could induce a heteroplasmic shift even when homoplasmic mtDNA variants are present, with the aim of attenuating or preventing the progression of pathological clinical phenotypes. In summary, mitochondrial transplantation is a challenging but potentially ground-breaking option for the treatment of various mitochondrial pathologies, although several questions remain to be addressed before its application in mitochondrial medicine.

## 1. Introduction

Mitochondria are cytoplasmic double-membrane organelles defined as eukaryotic cells’ powerhouses due to their involvement in the cellular bioenergetics. In particular, mitochondrial synthesis of adenosine triphosphate (ATP) is associated with the functionality of aerobic oxidative phosphorylation (OXPHOS) [1]. In addition to energy production, mitochondria play many critical roles in cellular function and signalling, including fatty acid biosynthesis, calcium homeostasis, reactive oxygen species production, cell survival, proliferation, apoptosis, autophagy, stem cell differentiation, and regulation of the immune response [2,3]. The complex functionality of mitochondria in cell biology is supported by their morphological and ultrastructural plasticity. Under physiological conditions, they form a dynamic and highly interconnected tubular network [4].

Due to their endosymbiotic origin [5], mitochondria have their own circular double-stranded DNA (mtDNA) that is transcribed and replicated following principles that are different from those regulating nuclear DNA (nDNA) [6]. Furthermore, unlike nDNA, mtDNA is characterised by maternal uniparental inheritance [7].

Human mtDNA is 16.6 kb long and comprises 37 genes, of which only 13 code for subunits of complexes I, III, IV, and ATP synthase. The remaining genes code for 2 ribosomal RNAs and 22 transfer RNAs, both of which are necessary for the protein synthesis machinery of mitochondria [8]. Nuclear genes encode the remaining proteins involved in OXPHOS and other enzymes required for mtDNA replication, repair, transcription, and translation [9]. Each individual mammalian cell contains hundreds or thousands of mitochondria, which in turn each contain between 1 and 15 molecules of mtDNA [10]. The ratio of mtDNA molecules/cell varies depending on cell and tissue types [11]. The mtDNA molecules—about 5 μm long—are highly compacted by non-histone proteins in aggregates (or spherical bodies) called nucleoids within the mitochondrial matrix [12]. In addition, mtDNA is characterised by the absence of protective histone proteins and a high exposure to oxygen free radicals (ROS) due to its proximity to OXPHOS sites [13]. The mtDNA repair processes, compared to nDNA, are less diversified, being primarily mediated by the base excision repair pathway [14]. Despite the high degree of conservation in animals, it is estimated that the mitochondrial genome evolves 10–20 times faster than nDNA, exhibiting heterogeneity in the nucleotide sequence [15]. The less efficient damage repair system, combined with continuous and random replication of mtDNA molecules, is probably the reason for the high mutation rate of the mitochondrial genome [16].

In the context of these random mutations, mitochondria exhibit polyplasmy, and the mtDNA genotype can result either from a single type of mtDNA (homoplasmy) or from the coexistence of several mtDNA haplotypes in different amounts (heteroplasmy) [17]. If the proportion of mutant molecules in the mitochondria exceeds a certain threshold, mitochondrial diseases can occur (threshold effect) [18].

In 1988, two research teams independently identified the first pathogenic mtDNA mutations: (1) a homoplasmic point mutation in MT-ND4 associated with Leber hereditary optic neuropathy (LHON) [19] and (2) large heteroplasmic deletions associated with mitochondrial myopathy [20]. Since then, the number of discovered mutations has increased significantly. To date, mitochondrial genome mutations are classified into two main groups associated with well-defined clinical syndromes: (1) large-scale rearrangements and (2) inherited point mutations [21].

Pathogenic mutations in nDNA or mtDNA cause mitochondrial diseases (MDs), a group of inherited genetic diseases characterized by impaired OXPHOS. MDs manifest as extremely heterogeneous, multisystemic diseases, that occur predominantly in highly OXPHOS-dependent tissues, such as the brain, skeletal muscle, heart, and the eye. However, any tissue can be damaged, and the disease can begin at any age, making MD diagnosis very difficult and leading to underestimation of their prevalence [22]. In MDs, a genetic defect can occur at either the mtDNA or nDNA level [23]. More specifically, the mitochondriopathies resulting from mtDNA mutations consist of: (1) large-scale mtDNA rearrangements with three primary clinical phenotypes—Kearns–Sayre syndrome (KSS, OMIM #530000), sporadic progressive external ophthalmoplegia (PEO, OMIM #157640), and Pearson syndrome (PS, OMIM #557000); (2) diseases caused by heteroplasmic point mutations of mtDNA, of which the most frequent phenotypes are: mitochondrial encephalomyopathy with lactic acidosis and stroke-like episodes (MELAS, OMIM #540000), myoclonic epilepsy with ragged red fibres (MERRF, OMIM #545000), neuropathy, ataxia, and retinitis pigmentosa (NARP, OMIM #551500); and (3) diseases caused by homoplasmic point mutations of mtDNA, such as Leber hereditary optic neuropathy (LHON, OMIM #535000) and mitochondrial nonsyndromic sensorineural hearing loss with susceptibility to exposure to aminoglycosides (SNHL, OMIM #500008).

Diagnosis, treatment, and development of effective therapies have been severely limited by the variability of MDs’ clinical expression and genetic heterogeneity. Currently, most therapies are aimed only at relieving symptoms [24].

In this manuscript, we will review current knowledge and advances in the field of mitochondrial transplantation to explore its potential as therapy for MDs and further discuss challenges and future perspectives.

## 2. Treatments for Mitochondrial Diseases

Due to recent developments in understanding the pathogenic mechanisms underlying mitochondrial disorders, various therapeutic approaches of “mitochondrial medicine” aiming to rescue wild-type phenotypes and prevent progression have been developed for MDs [25]. However, the development of efficient treatments has been severely hampered by the enormously variegate phenotype–genotype correlation that is typical of MDs. Since most of them lack an approved cure and available therapies are geared toward alleviating symptoms [26], there is urgent need to develop novel generalised or personalised approaches. To achieve this, different therapeutic strategies have been developed, which can be classified into two main groups: (1) pharmacological and metabolic approaches and (2) molecular approaches (Figure 1).

### 2.1. Pharmacological and Metabolic Approaches

Most of the pharmacological agents aim to reduce symptoms, particularly exercise intolerance and fatigue, and slow down the progression of the disease. Traditional treatments attempt to improve the function of the electron transport chain (ETC). In particular, several vitamins and cofactors are used to: (1) increase electron flux through the respiratory chain (CoQ10, riboflavin); (2) act as antioxidants (idebenone, alpha-lipoic acid, vitamins C and E), and/or increase ETC substrate availability (dichloroacetate and thiamine); and (3) feed the mitochondria (L-carnitine) [27]. However, in recent years, emerging experimental strategies have been proposed which are focused on: (1) stimulating mitochondrial biogenesis, (2) modulating mitophagy and mitochondrial dynamics, (3) bypassing OXPHOS deficits, (4) using mitochondrial replacement therapy (MRT), and (5) treating chronic hypoxia [24].

### 2.2. Molecular Approaches

Molecular strategies aim to override or correct genetic alteration to mitigate the consequent mitochondrial deficits. The nDNA-associated MDs can benefit from genetic approaches based on the targeted re-expression of the wild-type gene and from the genome editing techniques based on clustered regularly interspaced palindromic repeat/Cas9 (CRISPR/Cas9) technology, zinc finger nucleases (ZFNs), or transcription-activator-like effector nucleases (TALENs) [28]. However, mtDNA-related MDs must be treated with alternative methods. Indeed, unlike nDNA, not only is mtDNA not easily accessible due to the mitochondrial double membrane structure, but it is also present in multiple copies within the organelle [29]. Thus, the delivery of nucleic acids to mitochondria is challenging.

Three main experimental approaches have been developed to overcome this issue: (1) allotopic gene therapy, (2) mtDNA-selective endonuclease-based strategy, and (3) mitochondrial genome editing [28]. The first approach aims to re-express the mitochondrial gene at the nuclear level—followed by the import of the gene-encoded protein into the organelle, where it is supposed to function properly—by adding a mitochondrial-targeting sequence (MTS) at the N-terminus. Although clinical trials began in LHON patients, several challenges must be overcome to increase the successful application of this treatment [30]. The second approach exploits the observation that a break in the DNA double-strand leads to rapid degradation of the mtDNA molecule. Consequently, the selective elimination of mutated mtDNA *via* specific endonucleases might promote a heteroplasmic shift towards wild-type mtDNA copies within the organelle [31,32,33]. However, the strategy can only allow the potential treatment of heteroplasmic conditions. The third approach is based on two recent mitochondrial base editor technologies that are potentially able to correct homoplasmic single-nucleotide changes [33]. On the one hand, the DddA-derived cytosine base editor (DdCBE), which catalyses the conversions of C-to-T (or G-to-A) in human mtDNA [34]. On the other hand, transcription-activator-like effector (TALE)-linked deaminases (TALEDs) introduce targeted A-to-G editing in human mtDNA [35]. However, these approaches are still under development, and further studies are needed to verify their efficiency and safety in vivo.

## 3. Intercellular Transfer of Mitochondria

Cell-to-cell mitochondria transfer is considered a type of intercellular interaction that occurs physiologically in organisms [36]. Our knowledge of mitochondrial cell communication was enhanced by two experiments performed in the early 2000s: (1) Rustom et al. discovered cell-to-cell migration of organelles *via* nanotubular structures [37]; (2) Spees et al. demonstrated the natural transfer of wild-type mitochondria from mesenchymal stem cells (MSCs) into parenchymal cells displaying mitochondrial dysfunction [38]. Since then, a considerable body of data suggests that mitochondria and their components can be actively released into the extracellular space and transferred between cells under both healthy and pathological conditions [39,40]. Accordingly, knowledge of mitochondrial function is shifting from a microscale (single cell) to a macroscale cellular perspective, drawing attention to communication between cells through exchange of mitochondria.

The biological role of this interaction is still controversial and depends on (among other things) the metabolic state of the cells. In fact, mitochondria released from a donor cell might be degraded or integrated by a recipient cell [41]. According to some studies, transmitophagy processes are responsible for exogenous mitochondria degradation [42]: (1) Davis et al. show that the mitopsosis [43] of dysfunctional mitochondria from optic nerve neurons occurs in adjacent glial cells [44]; (2) Phinney et al. suggest the selective discharge of depolarized mitochondria from MSCs to macrophages. Importantly, the authors unexpectedly noted that the recipient macrophages integrate exogenous organelles in order to enhance their own bioenergetics [45].

In line with this, other research papers point out that damaged cells can also take up exogenous functional mitochondria from a donor cell and integrate them into their endogenous mitochondrial networks to improve biological processes [46]. This seems to be the case for: (1) astrocytes transferring mitochondria to neurons during focal cerebral ischaemia [47], (2) endothelial progenitor cells protecting the brain endothelium during stroke and brain injury [48], and (3) bone-marrow-derived mesenchymal stromal cells (BM-MSCs) providing mitochondria to diverse cell types in various conditions [49,50]. In short, intercellular communication between recipient and donor cells can generate what Liu et al. refer to as the “find me” and “save me” connection signal, a specific mechanism that occurs in damaged tissues [51].

In addition to the well-characterized apoptotic processes, other biochemical events have been observed in mitochondria under stress conditions. It could be that mtDNA escapes into the cytosol *via* the Bak/Bax permeabilization pathway, where it is identified by various pattern recognition receptors (PRRs) as a “foreign” damage-associated molecular pattern (DAMP). This process leads to the activation of the cGAS-STING1-TBK1-IRF3 cascade, which stimulates the expression of type I interferon and proinflammatory cytokine (IL-1, IL,4, IL-10 and TNFα) genes [52]. The mtDNA can also bind endosomal toll-like receptor 9 (TLR9) to boost a cellular proinflammatory response [53]. Given the cellular connections, one could boldly hypothesize that the proinflammatory response may also trigger a chain reaction involving the neighbouring cells. According to this view, a cell might both help the dysfunctional neighbouring cell to eliminate dysfunctional organelles and restore the pool of healthy mitochondria [54,55]. In view of the above, cell-to-cell mitochondrial transfer seems to be essential for the maintenance and restoration of the organism’s homeostasis [56].

### 3.1. Mechanisms of Mitochondria Intercellular Transfer

The molecular mechanisms responsible for mitochondrial transfer are far from being clearly and completely understood. The best-known molecular mechanisms for intercellular transfer of mitochondria are: (1) tunnelling nanotubes (TNTs) and (2) extracellular vesicles (EVs) (Figure 2).

#### 3.1.1. Transfer *via* TNTs

In 2004, Rustom and collaborators identified TNTs as distinct structures for intercellular communication while working on PC12 rat pheochromocytoma cell culture [37]. TNTs are plasma membrane-derived tubular cytoplasmic extensions of 50–1500 nm in width and 5–120 in length [57,58]. These dimensions facilitate unidirectional or reciprocal movement of signalling molecules and cellular components, promoting long-distance cell-to-cell interconnection [59]. Different cell types have been reported to form TNTs with each other in both physiological and pathological states [60]. TNTs have an F-actin-based structure and rely on adaptor–motor proteins to mediate cargo transfer [61].

It is possible to speculate that mitochondrial transfer between TNT-linked cells shares similar mechanisms with Miro1-mediated axonal transport of mitochondria into neurons. Indeed, Miro 1 (the outer mitochondrial membrane (OMM) Rho-GTPase Ca2+-dependent protein) appears to play a central role in regulating mitochondrial mobility along the TNTs [62]. In neurons, Miro1 acts as a mitochondria-loaded vehicle that interacts with OMM mitofusins and the molecular motor kinesin-1 *via* the Milton adapter protein (TRAK1/2). In this way, mitochondria move along the tracks of parallel polarized microtubule array [63]. The formation of TNTs is driven by the F-actin cytoskeleton-remodelling-based projection system. Three steps are involved in this mechanism: the development of a membrane protrusion, its expansion, and its fusion with the target cell membrane [64]. Currently, many proteins and pathways related to nanotubes formation have been identified [60,65].

#### 3.1.2. Transfer *via* EVs

Another system for cell-to-cell transfer of mitochondria is mediated by EVs. EVs were identified in 1971 by Aaronson S. et al., who discovered that the freshwater phytoflagellate Ochromonas danica produces membranous extracellular macromolecules [66]. Since then, a large number of reports in the last three decades, have suggested EVs’ involvement in intercellular signalling [67,68]. EVs are nano-sized circular phospholipid structures that are released into the extracellular space by almost all the cell types under both physiological and pathological conditions [69]. EVs are designated as biological cargoes because they carry and deliver a variety of bioactive molecules such as proteins, mRNAs, miRNAs, mtDNA, lipids (including cholesterol and cytokines), mitochondria, and their components [70,71]. They are classified in three general classes according to their nano-diameter and biogenesis: (1) exosomes (30–150 nm), which are of endocytic origin through inward budding of endosomal membrane; (2) microvesicles (150–1000 nm), which are formed by outward budding of the plasma membrane; and (3) apoptotic bodies (>1 μm), which are released by apoptotic cells [69].

Due to exosomes’ limited size, only microvesicles (eMVs) might contain intact mitochondria and contribute to mitochondrial intercellular transfer [72,73]. Indeed, recent papers have indicated horizontal mitochondrial transfer from donor cells to host cells through eMVs: (1) Islam et al. showed that bone-marrow-derived stromal cells (BMSC) supply alveolar epithelial cells with functional mitochondria *via* eMVs in a Cx43-dependent process [74]; (2) Phinney et al. reported that MSCs deliver eMVs with depolarized mitochondria to macrophages under pathological conditions [45]; (3) Hough et al. showed that airway myeloid-derived regulatory cells (MDRCs) secrete eMVs, which transfer mitochondria to T cells in a model of allergic airway disease [75]; (4) Davis et al. discovered that retinal ganglion cells can release mitochondria-rich EVs, which are captured by astrocytes in the optic nerve papilla [44]; and (5) Hayakawa et al. found that astrocytes release mitochondrial EVs that are captured by damaged neurons to support neuronal survival [76].

On the one hand, the eMV-mediated mitochondrial intercellular transfer mechanism is regulated by the NAD+/CD38/cADPR/Ca2+ pathway that activate the exocyst complex, leading to the formation and release of vesicles [36]. In addition, the loading of mitochondria into EVs may depend on optic atrophy 1 (OPA1) and sorting nexin 9 (Snx9) proteins, although the precise process is unknown [77].

On the other hand, anchorage on recipient cells is mediated by integrins localized on the eMVs surface [78]. After attachment to recipient cells, EVs can either immediately fuse with the membrane of the host cell or be internalized by a variety of mechanisms, such as phagocytosis, micropinocytosis, and clathrin-dependent endocytosis [79,80].

Overall, these finding suggest that cell-to-cell mitochondrial transfer pathway paves the way for a novel approach to mitochondrial transplantation for the treatment of different disorders.

## 4. Mitochondrial Transplantation (MT)

As further expansion in the field of innovative therapeutic strategies for mitochondrial disorders, two scientific evidences supported the idea of testing mitotherapy based on mitochondria delivery methods [81]. On one hand, according to the endosymbiotic theory, α-proteobacteria-derived mitochondria could be integrated into the host cell’s mitochondrial network [82]. On the other hand, evidence for cell-to-cell mitochondrial transfer suggested that mitochondria retain their bacterial ability to enter cells [83]. Many recent procedures of mitochondria transplantation rely on the transfer of exogenous functional mitochondria to injured cells to finally restore mitochondrial deficiency in several diseases [84].

### 4.1. In Vitro Methods for MT

Mitochondrial transplantation (MT) is an innovative technique based on the possibility of introducing healthy exogenous mitochondria into dysfunctional cells or tissues to modulate mitochondrial function [85]. The concept of “mitochondrial transplantation” was developed in 1982 by the pioneering study of Clark and Shay. These researchers purified mitochondria from donor human fibroblasts affected by mtDNA mutations, which induced chloramphenicol (CAP) and efrapeptin (EF) resistance (Figure 3A). Then, they transferred the organelles to antibiotics sensitive dysfunctional human recipient cells *via* simple coincubation method, and they were able to produce antibiotics-resistant vital mammalian cells, denominated “mitochondrial transformants” [86]. They also showed that antibiotics resistance could not be transferred from mouse to human cells. This mitochondrial transformation experiment led to some basic conclusions: (1) isolated mitochondria retain the ability to invade host cells through endocytosis mechanisms; (2) mtDNA from donor cells could be integrated into recipient cells, allowing them to transfer genetic material and induce functional changes; (3) integration of exogenous mitochondria from different species into the recipient mitochondrial network could potentially be limited by species barrier; and (4) cells might have variable mitochondrial uptake capacity according to their physiological state [87].

In 1988, King and Attardi confirmed that mitochondrial transformation is possible. They proposed the first precise MT technology for the transfer of healthy mitochondria in vitro. The researchers demonstrated that antibiotic resistance could be successfully generated in sensitive, partially mtDNA-depleted 143B human osteosarcoma cells through direct microinjection of human mitochondria isolated from CAP-resistant cell lines. They proved the retention of exogenous mtDNA copies in host cells from six to ten weeks after transplantation *via* the analysis of multiple mtDNA polymorphisms [88]. Although very accurate, the injection method was less efficient than Clark and Shay’s coincubation protocol because of the limited number of target cells in each transplantation experiment and the damage to the recipient cell. These preliminary data suggested that the uptake of a few healthy mitochondria could very rapidly repopulate a mtDNA depleted cells. In 1997, Pinkert et al. successfully microinjected mice liver mitochondria into mice zygotes, which incorporated a high rate of foreign mitochondria [89]. These results represented an initial step in the development of mitochondrial replacement therapy (MRT) techniques, which in MDs could prevent the transmission of mutant mtDNA to the foetus [90] (Figure 3B).

However, only in early 2000s, with the discovery of cell-to-cell transfer of mitochondria, did the real therapeutic potential of MT come to light. Since then, researchers’ interest triggered several MT studies on coincubation methods to investigate mitochondrial uptake in vitro [91,92]. The development of two strategies proved essential to assess the validity of MT: (1) advances in confocal microscopy technology [93] and (2) availability of Rho^0^ cells, completely deprived of mtDNA by ethidium bromide treatment [94], which are suitable models for proving mtDNA transfer [95]. Several MT approaches have been developed to improve in vitro transfer using chemical mediators. In 2017, Chang and his research group implemented Clark and Shay’s technique by using a penetrating peptide (Pep-1) conjugated with mitochondria to induce pores in the host cell’s plasma membrane to facilitate mitochondrial transfer. As a result, the conjugation of mitochondria isolated from human osteosarcoma 143B cybrid donor cells with Pep-1 promoted the mitochondrial internalization by a MELAS cybrid model (Figure 3C). The confocal microscopy analysis showed that the exogenous mitochondria colocalized with host cells mitochondria and favoured the recovery of the mitochondrial membrane potential [96]. In neonatal rat cardiomyocytes (NRCMs), Maeda et al. used transactivators of transcription dextran complexes (TATdextran) to increase cellular internalization of exogenous mitochondria and improve mitochondrial recipient function [97].

Remarkably, Caicedo et al. designed a protocol called MitoCeption based on the application of centripetal force to directly transfer mitochondria, isolated from MSCs, to human breast cancer MDA-MB-231 cells. In this protocol, mitochondria suspension is slowly added to the recipient cells in a Petri dish, which is then centrifuged and placed in the cell incubator facilitating the transplantation. The confocal imaging, fluorescence-activated cell sorting (FACS), and mitochondrial DNA analysis demonstrated the successful dose-dependent transfer of exogenous mitochondria to cancer cells [98]. Later, Cabrera et al. proposed a new transplantation protocol to simplify the MitoCeption procedure. Specifically, mitochondria are directly added into a microcentrifuge tube, where the recipient cells are in suspension. Following this, the tubes are centrifuged (Figure 3D). The novel protocol appears to be quicker and successful [99].

This procedure allowed for an allogenic transplant in vitro, repairing the metabolic activity, mitochondrial mass, and mtDNA sequence stability in UV-damaged peripheral blood mononuclear cells (PBMCs). In 2018, Kim’s research lab proposed an easy and quick centrifugation method without additional incubation steps. They first isolated mitochondria from human-umbilical-cord-derived MSCs *via* differential centrifugation, then transferred them into both L6 cells and human umbilical cord Rho^0^ cells *via* centrifugation. At the end, in the recipient cells, they observed an increase in ATP production, mitochondrial membrane potential, and oxygen consumption levels and a decrease in ROS [100]. According to Kim’s results, the application of an outer physical stimulus in MT experiments, such as centripetal force, appears to be successful regardless of cells’ types and physiological states. However, two critical issues must be considered. On one hand, the possible mechanical damage to the recipient cells’ plasma membrane. On the other hand, such a strategy cannot be performed in vivo. Here, the considerable therapeutic potential of stem cells becomes significant. As a matter of fact, stem cells from patients might be transplanted ex vivo and introduced back in an organism’s injured area to reprogram and/or repair its metabolism [55].

Such an attractive approach pushed researchers to explore novel physical-principle-based MT techniques [101]. In 2016, Wu et al. utilised a photothermal nanoblade to deliver mitochondria isolated from MDA-MB-453 cells into 143B- Rho0 cells [102] (Figure 3E). In detail, the photothermal nanoblade techniques causes the opening of the plasma membrane *via* laser pulses and the delivery of mitochondria using a titanium-coated borosilicate glass micropipette ~5 mm in length and with a tip ~3 μm in diameter using a fluid pump. Although the respiration of 143B- Rho^0^ cells seemed restored, only a few cells per hour could be transplanted in each experiment. Thus, the researchers adjusted the protocol with a biophotonic-laser-assisted surgery tool (BLAST) for the rapid massively parallel delivery of micron-sized large cargo into recipient cells [103].

In parallel, Macheiner’s groups proposed a novel mitochondrial transfer method, called magneto-mitotransfer, which employs an anti-TOM22 magnetic bead to labelled mitochondria and carries them into host cells with the support of a magnetic plate (Figure 3F). They reported the rapid autologous MT of human fibroblasts MRC-5 and the related functional changes. Moreover, in 2021, Sarcel et al. developed “MitoPunch”, a pressure-driven MT device to transfer mitochondria to recipient cells seeded on a porous polyester membrane (Figure 3G). They show the transfer of mitochondria isolated from HEK293T cells to 143B-Rho^0^ cells and the host retention of exogenous mtDNA copies [104].

Overall, the success of MT seems to be influenced by several factors: (1) the method of isolating mitochondria, which stresses organelles and affects their viability; (2) the selection of the donor cell line, with stem cells being preferred; (3) the amount of mitochondria to be transplanted; and (4) the delivery method [105]. The need to deliver intact and functional mitochondria has therefore led researchers to develop new tools.

In this scenario, Gäbelein’s group recently developed a single-cell technology called FluidFM to transplant mitochondria between living cells without compromising mitochondrial and cellular integrity or viability. FluidFM combines atomic force microscopy, optical microscopy, and nanofluidics. This procedure uses specific probes that permit minimally invasive access to cells and fluid flow to extract and inject mitochondria (Figure 3H). However, the technique is still not appropriate for use in vivo [106].

For in vitro and in vivo applications, the use of specific physiological vehicles with low immunogenicity and toxicity, such as extracellular vesicles (EVs), improved the preservation of mitochondria integrity when delivered (Table 1). Two main studies explored the novel EV-mediated MT method (Figure 3I). In 2021, Ikeda et al. isolated EVs from human-induced pluripotent stem cell (iPSC)-derived cardiomyocytes (iCMs) and coincubated them with recipient iCMs damaged by hypoxia. The confocal microscopy analysis indicated that exogenous mitochondria were transferred into host cells and fused with their endogenous mitochondrial networks, leading to a significant increase in ATP generation and an improvement in contractility [107]. In the same year, Peruzzotti-Jametti et al. investigated the hypothesis that neural stem cells (NSCs) release and transport functioning mitochondria through EVs into host cells to modulate mitochondrial activity. First, they verified whether the EVs were loaded with mitochondria *via* microscopic and functional analyses. Then, they incubated them with mtDNA-depleted L929 Rho^0^ cells. Mitochondrial function was then assessed in the recipient cells [71].

### 4.2. In Vivo Methods for MT

In addition to the reported in vitro MT studies, mitochondria can also be transplanted directly into animal models. In 2009, McCully et al. [108] presented the first in vivo MT study in an ischaemia–reperfusion heart model using an allogenic transplant in a rabbit. They isolated functional mitochondria from the left ventricular tissue of a healthy animal and injected them into the cardiac ischemic area of a different rabbit just before reperfusion. The study highlights the potential of MT to improve functional recovery of the heart and survival of cardiomyocytes after ischaemia. In addition, cardiomyocytes could uptake exogenous mitochondria just 2 h after injection. Further studies demonstrated that MT significantly reduces cardiac ischaemia markers (creatine kinase-MB and cardiac Troponin I) and the apoptosis protein Caspase-3, as well as infarct size [109]. Next, Cowan et al. performed two in vivo MT procedures: (1) direct injection of mitochondria into rabbit myocardial ischemic zone and (2) vascular perfusion of mitochondria through the coronary artery. Analysis of the results demonstrated that direct mitochondria injection proved to be more efficient in cardiac protection [110]. To summarize, direct mitochondrial transplantation appears to be the most promising way to modulate cardiomyocytes metabolism. Since then, several studies in vivo have been performed in different animal tissues, such as liver, lung, and brain [111] (Table 2). Worth to mention, two studies demonstrate the feasibility of MT on patients with ischaemia–reperfusion injury and single large-scale mtDNA deletion syndromes (SLSMDs).

In the first case, autologous mitochondria were isolated from a piece of healthy rectus abdominis muscle and direct injected to the myocardium of five paediatric patients who required central extracorporeal membrane oxygenation (ECMO) support for ischaemia–reperfusion dysfunction. The data show improvement in ventricular function within several days after treatment, but future studies are necessary to demonstrate the efficiency of the strategy [112].

Recently, in a compassionate use study, a mitochondrial augmentation therapy (MAT) approach has been applied on six patients with SLSMDs [160]. MAT is a cell technology platform in which autologous hematopoietic stem and progenitor cells (HSPCs) are augmented ex vivo with mitochondria obtained from donor cells or tissue. HSPCs have been shown to migrate to distal tissues and to abrogate disease-related deterioration of mitochondrial dysfunctions or metabolic disease. In this case, CD34+ cells were collected from a patient with a non-inherited primary mitochondrial disease, and PBMCs were collected from the maternal donor. Using a coincubation method, healthy mitochondria isolated from maternal PBMCs were transplanted into the patients’ CD34+ cells, which in turn were intravenously reinfused into the patients. MAT resulted in decreased heteroplasmy of mtDNA deletion, increased mtDNA levels, and improved ATP content in peripheral blood mononuclear cells in four out of six patients, as well as improved muscle strength and endurance in two individuals. Despite these encouraging results in peripheral tissues, this strategy does not reach the brain, which is also affected in SLSMDs (Table 2). However, this approach is not applicable in the case of homoplasmic mutations or in the presence of high heteroplasmy levels.

Several data show that mitochondrial dysfunction plays a pivotal role in neurodegeneration. Indeed, cells of the nervous system require a considerable amount of energy to carry out synaptic transmission, neurogenesis, and neuronal differentiation [161]. Therefore, MT strategy has also been experimented upon in animal models to treat neurodegenerative diseases and to investigate mitochondria supply in the brain [162].

For example, Shi et al. reported that systemic injection of mitochondria isolated from human hepatoma cells (HepG2 cells) into the brain of PD mice mitigates the progression of PD by improving ETC function and decreasing ROS production [143]. To facilitate mitochondrial transplantation, in another paper, the authors injected PeP-1-conjugated mitochondria into the medial forebrain bundle (MFB) of 6-OHDA PD rats. As a result, neuronal mitochondrial function and dynamics improved in the substantia nigra (SN) [142].

Additionally, encouraging data for MT in the CNS came from experimental models of schizophrenia (SZ) and Alzheimer’s disease [162]. Dos-Santos et al. demonstrated the therapeutic potential of MT in the crush-based animal model of optic-nerve glaucoma. The group performed an intravitreal injection of liver-isolated mitochondria and showed that MT transiently protects RGCs and reduces oxidative stress [156].

It is worth noting that the route of administration is crucial to the effect of MT in vivo. Currently, four routes of administration are used: (1) intracerebral injection, an invasive and painful procedure; (2) intrathecal injection, an alternative to bypassing the blood–brain barrier (BBB); (3) systemic or intracarotid administration, a less invasive and safer route; and (4) intranasal administration, which very recently has been shown as a simple and effective delivery system [140,144,163].

There are two main limitations to consider when performing MT in the CNS. First, the BBB which protects the brain from compounds that are toxic to brain neurons [164]. Second, the immune response in the brain is particularly alarming, and mitochondria are strongly immunogenetic organelles [165,166]. Indeed, some studies have investigated the immune reaction after mitochondrial transplantation in different tissue in vivo, confirming that there is an activation of the immune system following injection of mitochondria [167,168]. To reduce the immunological risk associated with MT, it would be very useful to untwist the mechanism underlying an immune response during this procedure [111]. In summary, future research on in vivo MT should focus on bypassing the BBB and preventing harmful immune responses in the brain. The study by Pluchino’s group mentioned above has provided a strategic method in this direction using NSC-derived EVs [71]. Remarkably, the delivery through vesicles not only preserves mitochondrial integrity but also reduces the damage associated with the immune response. Indeed, EVs hiding mitochondria from microglia and macrophages would act as a Trojan Horse, thus reducing immune reactivity.

## 5. Challenges and Future Prospective

Although mitochondrial transplantation has emerged as a possible therapeutic strategy for MDs, there are still several challenges to master before it can be used in clinical applications. While much progress has been made concerning the delivery, the remaining and still unsolved issues include: (1) mitochondrial yield and purity, (2) mitochondrial long-term storage, and (3) transplant rejection [169].

The isolation/preservation of undamaged and coupled mitochondria is an essential point. Indeed, it is now well known that dysfunctional mitochondria trigger a systemic or tissue-specific inflammatory response through various mechanisms, such as activation of inflammasomes and release of DAMPs [170].

Several methods of isolating mitochondria rely on differential centrifugation, which allows a rapid preparation characterised by high yield and low purity [171,172]. Techniques such as density gradient centrifugation and affinity purification by magnetic beads were used to further purify mitochondria or to separate distinct populations of mitochondria [172,173]. However, unlike differential centrifugation, yield is lower and purity is higher in these procedures [174]. Furthermore, isolation methods are time-consuming, and this could reduce mitochondrial viability and transplantation success [175]. Therefore, it is very important to define a suitable and rapid isolation method for mitochondrial transplantation.

According to McCully, isolated mitochondria can remain active and coupled for approximately 1 h on ice, and storage after this time significantly affects transplantation efficiency [175]. This storage range limits the clinical applications of MT. Thus, the therapeutic uses of MT could be substantially broadened if isolated mitochondria could be employed as a storable preparation [169]. Currently, when isolated mitochondria are cryopreserved at −80 °C, impairment of OMM integrity occurs [111]. In order to overcome this critical aspect, several studies of mitochondrial long-term storage have been made using cryoprotectors, such as DMSO [176] and trehalose [177]. However, despite the preservation of the integrity, these methods cause a reduction in mitochondrial functionality. Therefore, it is crucial to develop a method of mitochondrial cryopreservation to maintain both their stability and bioenergetic capacity.

Lastly, transplant rejection might occur in vivo. As mentioned above, some studies report the increase of autoimmune and inflammation markers after allogenic mitochondrial injection in different tissue in vivo [168]. This response could lead to the elimination of externally provided mitochondria from the organism to maintain a homeostatic state [167]. Two approaches could bypass this problem: (1) autologous mitochondrial transplantation, which caused no significant increase in immunity markers in animal models [109]; (2) mitochondrial delivery *via* EVs, which have lower immunogenicity compared to isolated mitochondria [178].

Interestingly, the extrusion of mitochondrial components *via* the generation and release of mitochondrial-derived vesicles (MDVs) from both the outer and inner membranes of mitochondria has been recently described. MDVs are mainly engaged for mitochondrial quality control and transporting selected mitochondrial cargoes [179]. Depending on the mitochondrial stress level, the cargoes can have a different fate. It has been demonstrated that they can be delivered to the late endosome/multivesicular bodies for packaging into EVs, to lysosomes for degradation, and to peroxisomes [180]. MDVs from damaged mitochondria might carry mtDNA, which—once inside EVs—could be released to the extracellular space. Neighbouring cells might recognise the genetic material as DAMPs and activate proinflammatory pathways [181]. These aspects suggest that MDVs generation and trafficking is regulated in a different way under pathological conditions. Future research focussing on MDVs as possible therapeutic tools to deliver mtDNA will have to take into account all the above-mentioned crucial issues.

Moreover, it has recently been shown that the retention time of exogenous mtDNA in recipient cells does not last long [91,182]. Indeed, once in the recipient cell, exogenous mitochondria could be integrated into the endogenous mitochondrial network *via* the fusion pathway (MFN1/2 and OPA1) and/or undergo mitophagy [183]. Chang and colleagues reported the increase in mitochondrial-fusion-related protein expression and inhibition of fission after transplantation [184].

Fusion and fission are two conserved processes which control mitochondrial dynamics, playing a key role in the regulation of mitochondrial remodelling and cellular metabolism [185,186].

Fission is mediated by the interaction of a cytosolic GTPase, dynamin-related protein 1 (DRP1) with four OMM proteins (Fis1, MiD49, MiD51, Mff), resulting in the division of the mitochondrion into two similar organelles [187].

Fusion of the outer and inner mitochondrial membranes, on the other hand, requires the action of two distinct classes of integral DRPs, such as the mitofusins (MFN1/MFN2) located at the OMM level and optic atrophy protein 1 (OPA1), located in the intermembrane space [3].

Defects in both mitochondrial fission and fusion impair mitochondrial function and contribute to numerous pathological conditions. Collectively, mitochondrial dynamics evolve as a compensation mechanism to maintain heteroplasmic condition in cells. Indeed, when mutated mtDNA accumulates, fusion compensates mitochondrial function by mixing wild-type and defective mtDNA copies [188]. On the other side, fission antagonizes fusion processes and triggers segregation of damaged mitochondria for subsequent elimination by mitophagy [81].

Amongst several other challenges when transplanting mitochondria, it is crucial to better understand how mitochondrial fission and fusion are regulated in order to promote the integration of exogenous mtDNA copies into the host endogenous mitochondrial network.

On the other hand, the selective elimination of external mitochondria represents a barrier to organelles’ retention. Two reasons could lead to transplant rejection: (1) genome balance; and (2) haplotype matching between exogenous and endogenous mtDNA (for a comprehensive review, see [189]).

The concept of “genomic balance” refers to the relationship between mitochondrial and nuclear genomes to maintain cellular destiny and homeostasis [190,191,192,193]. Thus, the administration of exogenous mtDNA copies with different genetic profiles could disrupt the existing balance between mtDNA and nDNA in recipient cells. These mtDNA–nuclear mismatches may lead to complications in male mice [194]. At the same time, mtDNA–mtDNA interactions could be problematic. For example, in mice, the combination of two mtDNA haplotypes from the same subspecies resulted in physiological alterations [195]. Collectively, it could be that to restore homeostasis, either the host cell is reprogrammed or exogenous mitochondria are eliminated as a compensatory mechanism. Consequently, future investigations should focus on unravelling the mechanisms behind the mitonuclear and/or mtDNA–mtDNA interactions to better understand how these relations could affect mitochondria transplantation.

The transplantation of autologous mitochondria would prevent this mismatch as they are derived from the same patient. Conversely, this would not be appropriate for individuals with congenital mitochondrial disorders. To minimize mtDNA difference during allotransfer, it has been proposed that mitochondria derived from genetically close family members from the maternal lineage without the mutation or with very low heteroplasmy levels should be used [169]. A future solution may be the application of the in vitro mtDNA editing approaches combined with the possibility to generate induced pluripotent stem cells (iPSC) from patient somatic cells. In this scenario, patient-derived iPSC could be used tout court or differentiated in specific cell types, such as mesenchymal or neural precursor cells, in order to induce the reversion of the pathogenetic mutation outside of the body. Subsequently, after appropriate quality control and selection steps, these modified cells may be used as ex vivo donors for mitochondrial transplantation.

## 6. Conclusions

The main aim of this review is to investigate mitochondrial transplantation as a potential therapeutic strategy for MDs. Diagnosis, treatment, and the development of effective therapies are severely limited by the variability in MDs’ clinical expression. Unlike heteroplasmic conditions, MDs caused by homoplasmic mtDNA mutations do not benefit yet from advances in molecular approaches. Therefore, providing healthy mitochondria in the presence of dysfunctional mitochondria appears to be an attractive option, paving the way to novel, intriguing alternatives. Interest in mitochondrial transplantation was sparked by the discoveries of intercellular transfer of mitochondria in the early 2000s. Since then, several studies have been carried out to improve MT techniques and evaluate their validity in different disease models both in vitro and in vivo. The rationale of MT is to induce a heteroplasmic shift in a homoplasmic MDs condition to attenuate or prevent the progression of pathological phenotypes. Three aspects must be considered for a successful transplantation: (1) maintenance of mitochondrial function and integrity during transfer, (2) mitochondrial immunogenicity, and (3) metabolic/genetic interaction of exogenous mitochondria with mitochondria of the recipient cells. Once these conditions are met, mitochondrial transplantation could become a reality to overcome the limitations of currently available approaches, designed to correct specific mutations with the unprecedented advantage of having a broad applicability in various mitochondrial diseases.

## Figures and Tables

**Figure 1 ijms-24-01969-f001:**
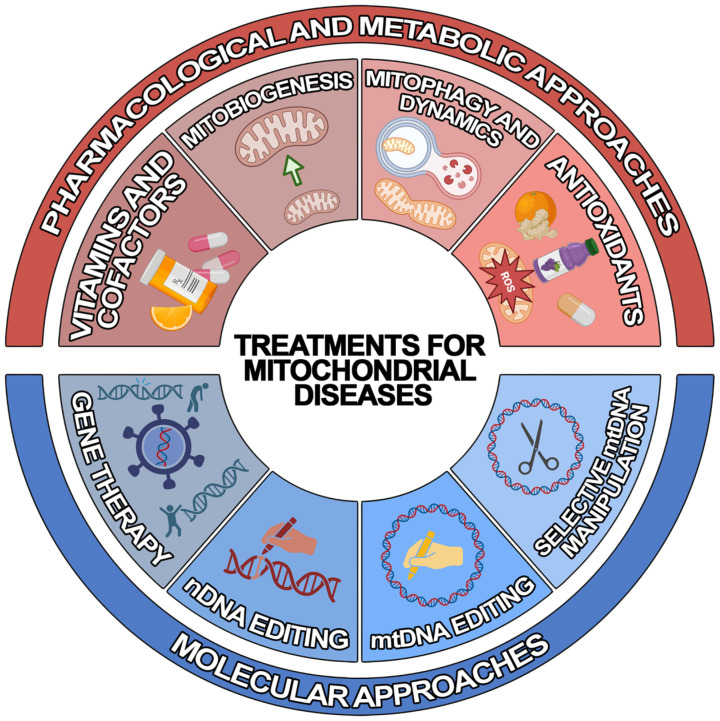
Schematic representation of the treatments for mitochondrial diseases. Illustrated are the current treatment strategies for mitochondrial diseases, which can be classified into pharmacological and metabolic (red) and molecular (blue) approaches.

**Figure 2 ijms-24-01969-f002:**
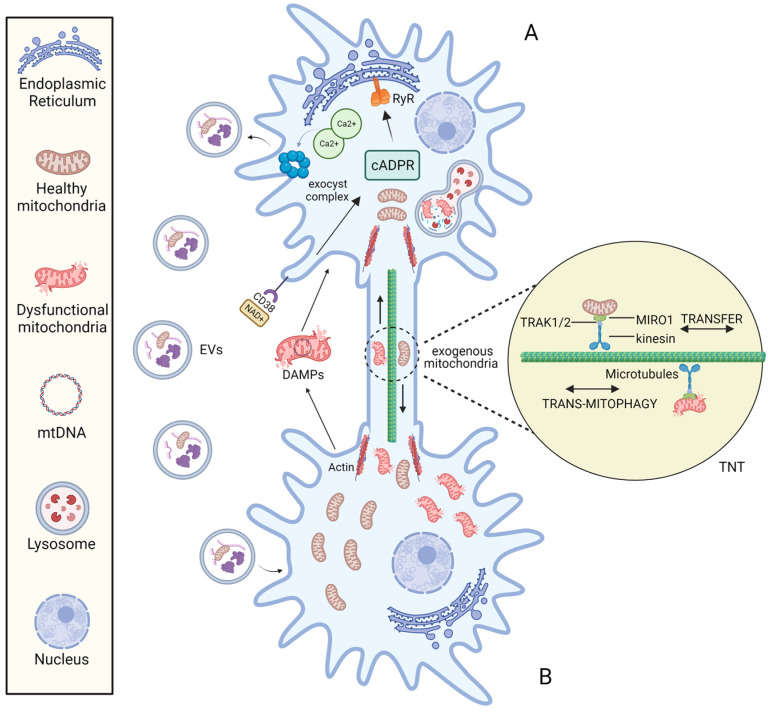
The mechanisms of intercellular mitochondrial transfer. Under the stimulation of stress conditions, mitochondria can be transported bidirectionally between cell A and B *via* the TNT structure and mitochondria-containing EVs. The formation of TNT is driven by the actin, and mitochondrial transfer between TNT-linked cells is regulated by Miro1. EV endocytosis is mediated by the NAD^+^/CD38/cADPR/Ca^2+^ pathway: Under stress, intracellular NAD+ increases and diffuses to the extracellular environment. CD38 then catalyses NAD^+^ to produce cADPR, a second messenger that acts on Ryanodine Receptor (RyR) on the Endoplasmic reticulum to induce the release of the intracellular Ca^2+^. Following the increase of cytoplasmic Ca^2+^ mediates the activation of exocyst complex, leading to the formation and release of vesicles.

**Figure 3 ijms-24-01969-f003:**
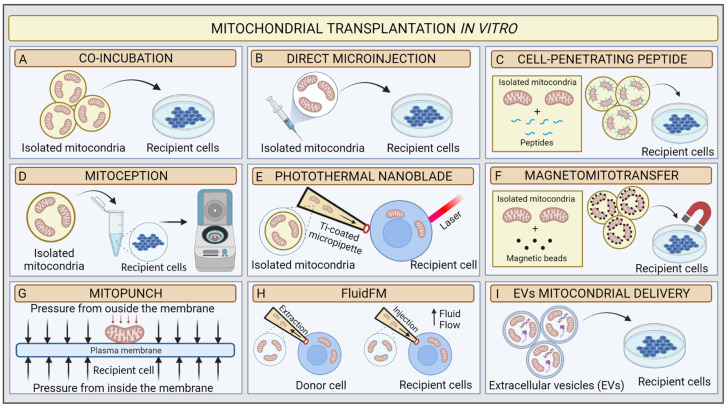
Schematic illustration of mitochondrial transplantation methods in vitro. (**A**) Coincubation, (**B**) direct microinjection, (**C**) cell-penetrating peptide mitochondrial delivery, (**D**) mitoception, (**E**) photothermal nanoblade, (**F**) magnetomitotransfer, (**G**) Mitopunch, (**H**) FluidFM, (**I**) EV mitochondrial delivery.

**Table 1 ijms-24-01969-t001:** A comparison of in vitro mitochondrial transplantation methods.

In Vitro MT Methods	Pros	Cons	Remarks
Co-incubation	Reduced manipulation Broad number of transplanted recipient cells Easy to realise	Low accuracy Dependence on physiological state and uptake capacity of recipient cells Mitochondria dose-dependent High risk of mitochondria damage	mtDNA retention up to 12 passages Moderate transfer Efficiency
Microinjection	Successful regardless the physiological state and uptake capacity of the target cells	Potentially harmful for the target Limited number of cells that can be transplanted per experiment High risk of mitochondria damage	mtDNA is retained from 6–10 weeks after treatment
Cell-penetrating peptide	Low manipulation Increased uptake capacity rate of target cells	Unknown effect of Pep-1 on mitochondrial function High risk of mitochondria damage	mtDNA is retained 11 days after Treatment
MitoCeption	Time saving Successful regardless the physiological state and uptake capacity of the target cells	High manipulation Potentially harmful for the target Mitochondria dose-dependent High risk of mitochondria damage	mtDNA retention not known Moderate transfer Efficiency
Photothermal nanoblade	Rapid massively delivery Very accurate	High manipulation Specific equipment High risk of mitochondria damaging Limited number of transplanted cells	Stable retained 2% transfer efficiency
Magnetomitotransfer	Rapid massively delivery Very accurate	Specific supplies High risk of mitochondria damage	mtDNA retention not known High transfer Efficiency
Mitopunch	Rapid massively delivery	Not suitable for all cell types (only those attaching PET filter) Dependent on nDNA-mtDNA mismatch High risk of mitochondria damage	Stable retention Moderate transfer Efficiency
FluidFM	Mitochondria and cellular integrity preservation Minimally invasive	Specific equipment High cost	High transfer efficiency mtDNA retention is not known
EVs mitochondrial delivery	Low manipulation Mitochondrial and cellular integrity preservation Easy to realise	Mitochondria-rich-EV isolation	mtDNA retention not known

**Table 2 ijms-24-01969-t002:** In vivo research reports of mitochondrial transplantation.

Targeted Organs	Species	Disease	Route of Administration	Studies Reference
Heart	Rabbits, pigs, rats, mice, piglets, humans	Heart regional/global ischaemia; heterotopic heart transplantation; right heart failure.	Local direct injection; intracoronary injection.	[107,108,109,110,112,113,114,115,116,117,118,119,120,121]
Liver	Rats, mice	Partial liver ischaemia; fatty liver; acetaminophen/carbon-tetrachloride-induced liver injury.	Intrasplenic injection; intravenously injection.	[122,123,124,125,126,127,128]
Lung	Rats, mice	Airway hyperresponsiveness; melanoma lung metastasis; acute lung ischaemia–reperfusion; pulmonary hypertension, experimental sepsis.	Intratracheally injection; intravenously injection; intracoronary injection; pulmonary artery injection.	[129,130,131,132,133]
Brain	Rats, mice, pigs	Stroke; Parkinson’s; schizophrenia; Alzheimer’s; age-associated cognitive decline, depression; spinal cord injury; optic nerve crush.	Intracerebral injection; systemic injection; intrathecal injection; intranasal injection; intracerebroventricular injection.	[71,134,135,136,137,138,139,140,141,142,143,144,145,146,147,148,149,150,151,152,153,154,155,156,157,158,159]
Blood	Human	Single large-scale mtDNA deletion syndromes.	Intravenous reinfusion of CD34^+^ ex vivo transplanted cells.	[160]

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
