# Peer review of "Mitochondrial Transplantation in Mitochondrial Medicine: Current Challenges and Future Perspectives"

_ijms, 2023, doi:10.3390/ijms24031969_

Round 1

Reviewer 1 Report

It is a quiet interesting story. I believe that mitochondrial diseases are very complex diseases and there is almost no cure for MDs. In other words, mitochondria medicine is an emerging and exciting field. I think that the strength of the manuscript is the description of several potential models and well organized/appropriated references for the story.

Author Response

We thank the reviewer for appreciating our manuscript

Reviewer 2 Report

This review article is well-written and recently published articles are cited. The figures are interesting and present informative data on the subject.

It would be better to add a paragraph on the mitochondrial dynamic (fission/fusion) and its impact on the efficacy of transplantation.

Please include a Table about clinical trials on mitochondrial transplantation therapy (in different diseases) for readers.

It is better to include sections for methods used for preserving and stability of mitochondria during and after transplantation.

How the host immune system responds to transplanted mitochondria.

It is important to highlight the challenges in mitochondrial transplantation in the clinic. 

Author Response

This review article is well-written and recently published articles are cited. The figures are interesting and present informative data on the subject.

It would be better to add a paragraph on the mitochondrial dynamic (fission/fusion) and its impact on the efficacy of transplantation.

Answer: according to this suggestion we discussed the implications of fission and fusion on efficacy of mitochondrial transplantation in section 5 "Challenges and future prospective".

Please include a Table about clinical trials on mitochondrial transplantation therapy (in different diseases) for readers.

Answer: considering that this review is specifically dedicated to the potential application of Mitochondrial Transplantation (MT) to mitochondrial disorders, we believe it would be misleading to include a table containing the application of MT in different diseases. We improved table 1 (now table 2 after the revision requested by another reviewer) by including the only clinical trial now available based on MT, carried out in the mitochondrial disorder Pearson.

It is better to include sections for methods used for preserving and stability of mitochondria during and after transplantation.

Answer: This aspect has been mentioned in the manuscript. Please refers to this paragraph: “According to McCully, isolated mitochondria can remain active and coupled for approximately 1 hour on ice, and storage after this time significantly affects transplantation efficiency [175]. This storage range limits the clinical applications of MT. Thus, the therapeutic uses of MT could be substantially broadened if isolated mitochondria could be employed as a storable preparation [169]. Currently, when isolated mitochondria are cryopreserved at -80 °C, impairment of OMM integrity occurs [111]. In order to over-come this critical aspect, several studies of mitochondrial long-term storage have been made using cryo-protectors, such as DMSO [176], and trehalose [177]. However, despite the preservation of the integrity, these methods cause a reduction of mitochondrial functionality. Therefore, it is crucial to develop a method of mitochondrial cryopreservation to maintain both their stability and bioenergetic capacity.”

How the host immune system responds to transplanted mitochondria.

Answer: Also this aspect has been mentioned in the manuscript. Please refers to this paragraph: “Last, but not least, transplant rejection might occur in vivo. As mentioned above, some studies report the increase of autoimmune and inflammation markers after allogenic mitochondrial injection in different tissue in vivo [168]. This response could lead to the elimination of externally provided mitochondria from the organism to maintain a homeostatic state [167]. Two approaches could bypass this problem: (1) autologous mitochondrial transplantation, which caused no significant increase of immunity markers in animal models [109]; (2) mitochondrial delivery via EVs, which have lower immunogenicity compared to isolated mitochondria [178].”

Moreover, we further discussed the issue on immune system response in the section “In vivo methods for MT”. Please refers to this paragraph: “There are two main limitations to consider when performing MT in the CNS. First, the BBB which protects the brain from compounds that are toxic to brain neurons [164]. Second, the immune response in the brain is particularly alarming and mitochondria are strongly immunogenetic organelles [165,166]. Indeed, some studies have investi-gated the immune reaction after mitochondrial transplantation in different tissue in vi-vo, confirming that there is an activation of the immune system following injection of mitochondria [167,168]. To reduce the immunological risk associated with MT, it would be very useful to untwist the mechanism underlying an immune response during this procedure [111]. In summary, future research on in vivo MT should focus on bypassing the BBB and preventing harmful immune responses in the brain. The study by Pluchi-no’s group mentioned above has provided a strategic method in this direction, using NSC-derived EVs [71]. Remarkably, the delivery through vesicles not only preserves mitochondrial integrity, but it also reduces the damage associated with the immune re-sponse. Indeed, EVs hiding mitochondria from microglia and macrophages, would act as a Trojan horse thus reducing immune reactivity.”

It is important to highlight the challenges in mitochondrial transplantation in the clinic.

Answer: We have described in details all the challenges connected to the application of MT in the clinic in the last chapter “5. Challenges and future prospective”. Specifically we discussed the following issues: the mitochondrial yield and purity; the mitochondrial long-term storage and preservation, the transplant rejection, the immune response, and the concept of “genomic balance”.

Reviewer 3 Report

In this manuscript, the authors introduced mitochondrial diseases (MDs), and treatments for MDs, and summarized the current knowledge about the mechanisms of intercellular transfer of mitochondria, and approaches for mitochondrial transplantation in vitro and in vivo in detail. The manuscript is very interesting and of significant contribution to this field.

My major concerns are below:

1.       According to this manuscript, MDs are defined as inherited genetic conditions characterized by pathogenic mutations in nuclear DNA (nDNA) or mitochondrial DNA (mtDNA). The “Introduction” and “Treatments for mitochondrial diseases” sections mainly focused on MDs, and the authors mentioned that the main aim of this review is to investigate mitochondrial transplantation as a potential therapeutic strategy for MDs. However, in the “Mitochondrial transplantation (MT) “section, in vivo methods for MT part, all the disease models referred, though with mitochondrial dysfunction or damage, are not MDs. Please revise to make the manuscript more consistent.

2.       I understand that this manuscript is focusing on the mechanisms and approaches to entire mitochondria transfer. Mitochondrial-derived vesicles (MDVs) are widely investigated recently and can serve as cargos to deliver mtDNA. MDVs are also expected as a promising therapeutic approach for MDs. Would you please discuss a little bit about MDVs in your manuscript?

3.       The mitochondrial transplantation doesn’t carry nDNA, will this limit the application of MT in nDNA-mutant-associated MDs?

4.       A table to summarize the approach, advantages, disadvantages, and limitations of different mitochondrial transplantation methods in vitro may help the reader to understand the characteristics of in vitro methods for MT better.

Minor suggestions:

1.       In line 364, “the application of an outer physical stimulus in TM experiments, such as”. What’s the full name of TM experiments? Or do you mean MT experiments?

2.       In line 396, “the success of MT seems to be influenced by several factors”. Would you please also include the method to deliver mitochondria, which is a pivotal factor.

Author Response

We thank the reviewer for the suggestions. Please find below a point-by-point answer to the raised questions.

My major concerns are below:

  1. According to this manuscript, MDs are defined as inherited genetic conditions characterized by pathogenic mutations in nuclear DNA (nDNA) or mitochondrial DNA (mtDNA). The “Introduction” and “Treatments for mitochondrial diseases” sections mainly focused on MDs, and the authors mentioned that the main aim of this review is to investigate mitochondrial transplantation as a potential therapeutic strategy for MDs. However, in the “Mitochondrial transplantation (MT) “section, in vivomethods for MT part, all the disease models referred, though with mitochondrial dysfunction or damage, are not MDs. Please revise to make the manuscript more consistent.

Answer: We thank the reviewer for this comment. At the moment, apart from a clinical trial recently published on Pearson syndrome, an MD disorder characterized by mtDNA macrodeletion, no MT based approaches in mitochondrial disorders have been reported in the literature. In this revised version of the manuscript we inserted, in the section “in vivo methods for MT”, two examples of MT in vivo, including the one performed in Pearson syndrome, that in the previous version, was mentioned in the section “Challenges and future prospective”. We also improved the table (now Table 2) accordingly.

  1. I understand that this manuscript is focusing on the mechanisms and approaches to entire mitochondria transfer. Mitochondrial-derived vesicles (MDVs) are widely investigated recently and can serve as cargos to deliver mtDNA. MDVs are also expected as a promising therapeutic approach for MDs. Would you please discuss a little bit about MDVs in your manuscript?

Answer: We added a sentence commenting MDVs in “Challenges and future prospective” and we cited supporting literature.

  1. The mitochondrial transplantation doesn’t carry nDNA, will this limit the application of MT in nDNA-mutant-associated MDs?

Answer: This won’t represent a limitation. MT can be applied also in nuclear DNA associated MDs, since the important aspect in this case is to increase the levels of “Functional Mitochondria” irrespective of the mtDNA, which is instead important in case of mtDNA related mitochondrial disorders.

  1. A table to summarize the approach, advantages, disadvantages, and limitations of different mitochondrial transplantation methods in vitromay help the reader to understand the characteristics of in vitro methods for MT better.

Answer: According to this reviewer’s suggestion, we included a new table (now Table 1) in the manuscript.

Minor suggestions:

  1. In line 364, “the application of an outer physical stimulus in TM experiments, such as”. What’s the full name of TM experiments? Or do you mean MT experiments?

Answer: Thank you for this note. We corrected accordingly.

  1. In line 396, “the success of MT seems to be influenced by several factors”. Would you please also include the method to deliver mitochondria, which is a pivotal factor.

Answer: We included the method to deliver mitochondria in the list.

Round 2

Reviewer 2 Report

It can be accepted

Reviewer 3 Report

Thanks so much for all the authors' effect. The manuscript sound great to me now.